# Proactive Use of a Human Milk Fat Modular in the Neonatal Intensive Care Unit: A Standardized Feeding Protocol

**DOI:** 10.3390/nu16081206

**Published:** 2024-04-18

**Authors:** Amanda Salley, Martin L. Lee

**Affiliations:** 1Hinsdale Hospital, UChicago Medicine AdventHealth Hinsdale Hospital, Hinsdale, IL 60521, USA; 2Prolacta Bioscience, City of Industry, CA 91746, USA; 3Department of Biostatistics, UCLA Fielding School of Public Health, Los Angeles, CA 90025, USA

**Keywords:** exclusive human milk diet, human milk fat modular, growth, standardized feeding protocol, NICU, cost savings

## Abstract

An exclusive human milk diet (EHMD) and standardized feeding protocols are two critical methods for safely feeding very low birth weight (VLBW) infants. Our institution initiated a standardized feeding protocol for all VLBW infants in 2018. In this protocol, a human milk fat modular was used only reactively when an infant had poor weight gain, fluid restriction, or hypoglycemia. As part of our NICU quality improvement program, internal utilization review data revealed a potential opportunity to improve growth and reduce costs. While maintaining the EHMD, a simple feeding guideline process change could provide cost savings without sacrificing caloric density or growth. We examined this process change in pre-post cohorts of VLBW infants. Methods: Our revised feeding protocol, established in October 2021, called for a human milk fat modular (Prolact CR) to be added to all infant feeding when parenteral nutrition (PN) and lipids were discontinued. The human milk fat modular concentration is 4 mL per 100 mL feed, providing approximately an additional 2 kcal/oz. We tracked data to compare (1) the use of the human milk fat modular, (2) the use of the human milk +8 fortifier, (3) overall growth before and after feeding protocol changes, and (4) cost differences between protocols. Results: Thirty-six VLBW infants were followed prospectively upon the introduction of the revised feeding protocol. In the revised era, the need for human milk +8 fortifier decreased from 43% to 14%. The decrease in the cost of a more costly fortifier provided a cost savings of USD 2967.78 on average per infant. Overall growth improved from birth to discharge, with severe malnutrition declining from 3.3% to 2.7% and moderate malnutrition declining from 37% to 8%. Conclusions: With the proactive use of a human milk fat modular in a standardized feeding protocol, our VLBW infants showed improved growth, lower malnutrition rates, and decreased use of higher caloric fortifiers.

## 1. Introduction

An exclusive human milk diet (EHMD) utilizing a mother’s own milk (MOM) with human milk-based fortifiers and the use of a standardized feeding protocol are two of the most evidence-based methods for decreasing necrotizing enterocolitis, improving growth, and decreasing overall co-morbidities in very low birth weight (VLBW) infants [1,2,3]. The American Academy of Pediatrics (AAP) and Surgeon General recommend the use of human milk for premature infants [3]. Huston et al. suggested that early fortification with an EHMD can improve growth and significantly decrease NEC (necrotizing enterocolitis).

The UChicago Medicine AdventHealth Hinsdale neonatal intensive care unit (NICU) initiated the use of an EHMD (Prolacta Bioscience, Inc., Duarte, CA, USA) in January 2016 for all VLBW infants. At the Level III NICU, an EHMD was provided to all infants born with £1500 g and/or £32 0/7 weeks at birth. In 2018, a standardized feeding protocol was implemented in an attempt to improve growth and maintain consistency within the NICU. After an internal review of these NICU’s VON (Vermont Oxford Network) data in 2019, it became apparent that NEC had risen beyond typical percentiles. Most infants affected were prenatally diagnosed with end diastolic flow, maternal abruption, metabolic acidosis at birth, and/or IUGR (intrauterine growth restriction). In 2020, an updated standard (Figure 1) and modified feeding protocol (Figure 2) were implemented to improve outcomes in our VLBWs.

Within the updated 2020 feeding protocol, two significant changes were made. All infants born at £1000 g followed a protocol in which feeds were fortified with Prolact +4 (Prolacta Bioscience, City of Industry, CA, USA) or RTF 24 kcal/oz when feeds were at trophic level, less than 20–30 mL/kg. All infants following the feeding protocols received fortification of Prolact +6 (Prolacta Bioscience, City of Industry) or RTF 26 kcal/oz when feeds reached 60 mL/kg. Early fortification supports optimal growth and allows improved nutrients to be administered enterally rather than parenterally [1,4,5]. The second notable change coincides with parenteral nutrition (PN) guidelines. The updated feeding protocol identifies when enteral feeds are included in the total fluid volume. Utilizing the appropriate feeding protocol with early fortification and PN guidelines [1,5], both metabolic acidosis and metabolic bone disease can be avoided [6,7].

In October 2021, a chart review was conducted to determine the success of the current feeding protocol. Through this review, it was determined that 77% of infants utilized a human milk fat modular, and 43% of infants required a human milk +8 fortifier in addition to the fat modular. As part of our NICU quality improvement program, internal utilization review data revealed a potential opportunity to improve growth and reduce costs. Our hypothesis was that proactive use of a human milk fat modular, given after providing appropriate enteral protein intake, could improve growth and decrease cost through less use of the more costly +8 human milk fortifier.

Based on previous studies, it is known that fat loss is high in human milk feedings given via tube feeding [8,9]. The Rogers et al. study resulted in 6 ± 2% loss of fat via gravity feeds, 13 ± 3% loss of fat via pumps, and 40 ± 3% loss of fat in continuous feeds [8]. Once enteral protein needs are met (3.5–4.5 g/kg/day), proactively utilizing a human milk fat modular might allow for the provision of sufficient nutrients for optimal growth without the need for a more costly fortifier. Knake et al. found that 73% of infants in their study required the use of a cream supplement for weight gain <15 g/kg/d [9]. Tabata et al. displayed benefits from a human milk-derived fortifier and a human milk fat modular to improve infant weight gain with bioactive elements from mother’s milk and increased fat delivery [10]. Hair et al. hypothesized that premature infants who receive an EHMD with a human milk fat modular would have weight gain at least as good as infants receiving a standard feeding regimen that consisted of MOM or donor HM with a human milk-derived fortifier [11]. Within this study, the authors found a significant enhancement in the growth of preterm infants who received a human milk fat modular in conjunction with an EHMD.

Proactive use of a human milk fat modular can meet the additional needs required by VLBW infants rather than waiting for reactive usage when growth remains poor. Utilizing an EHMD with reactive fat modular use, this Level III NICU was seeing moderate malnutrition indicators in 36.6% of infants (Table 1).

Following the chart review, the standardized feeding protocol adopted a more proactive use of the human milk fat modular (Figure 3).

## 2. Materials and Methods

In implementing the new standardized feeding protocol (Figure 3), the human milk fat modular (Prolact CR, Prolacta Bioscience, City of Industry) is added to all feedings at 110–120 mL/kg when PN/lipids are discontinued. Prolact CR (Prolacta Bioscience, City of Industry) is mixed at 4 mL per 100 mL feeding, providing an average of an additional 2 kcal/oz. Enteral feeds are given over 30 min, unless a physician order is provided for an increased duration. Although not a preferred practice, when continuous feeds are needed in Level III, Prolact CR (Prolacta Bioscience, City of Industry) is provided in the same ratio, divided, and given as a bolus every 4 h before a new feeding syringe is placed [8].

Data were collected by the neonatal dietitian to compare the use of the human milk fat modular, the use of the human milk +8 fortifier, and overall growth.

### 2.1. Inclusion/Exclusion Criteria

This quality improvement study was approved by the institutional review board for UChicago Medicine AdventHealth Hinsdale, a level III NICU in the west suburbs of Chicago. The data were obtained through a retrospective review (births from January 2021 to October 2021) and ongoing data collection post-protocol changes (births from November 2021 to June 2022). Infants utilizing the EHMD—all infants born at £1500 and/or £32 weeks—were included in the study. Group 1 was obtained from a retrospective review, and Group 2 was established following the timing of protocol changes. Infants with presumed milk protein allergy, necrotizing enterocolitis (NEC) or intestinal perforation, transportation to an outside hospital, or need for an early wean were excluded from data collection (Table 2). This quality improvement initiative was to evaluate infants who could stay on a standardized feeding guideline and did not require a modified approach.

### 2.2. Feeding Protocols

All infants were provided with an EHMD via a standardized feeding protocol (Figure 1). MOM was provided when available, or pasteurized donor human milk (DHM) was provided if the mother’s own milk was unavailable. Infants’ feeds were fortified with an exclusive human milk fortifier to 26 kcal/oz when feeds reached 60 mL/kg within the feeding protocol advancement. If an infant was experiencing poor weight gain, defined as <15 gm/kg/day, a human milk fat modular was added to feeds to provide an additional 2 kcal/oz. Furthermore, if weight velocity was still not meeting adequate levels (<15 g/kg/day), the human milk fortifier was further advanced to 28 kcal/oz, in addition to the human milk fat modular. Our goal growth velocity is 15–20 g/kg/day based on the most up-to-date literature for preterm infants <2 kg [12]. Group 1 was fed an EHMD with reactive use of a human milk fat modular when poor weight gain was seen over 3–4 days and further advanced to 28 kcal/oz if needed to support optimal weight gain [12].

In Group 2, all infants were provided an EHMD via the updated standardized feeding protocol (Figure 3) with the addition of prophylactic use of a human milk fat modular, Prolact CR (Prolacta Bioscience, City of Industry), when feeds were at 110–120 mL/kg. This addition of the human milk fat module coincided with the discontinuation of parenteral nutrition and lipids. The human milk fat modular was provided in the same ratio as Group 1. Infants were provided with 4 mL cream for every 100 mL of fortified feed. If an infant within Group 2 was seen to have poor weight gain (<15 g/kg/day), the human milk fortifier was advanced to 28 kcal/oz.

Group 1 and Group 2 were both weaned off an exclusive human milk diet when the infant was both 33 3/7 weeks and 1500 g. The wean occurred over a 4-day period and was complete when the infant was 34 0/7 weeks.

### 2.3. Data Collection

Data were collected by the neonatal dietitian from admission to discharge on all infants in Group 1 and Group 2. In addition to birth anthropometrics and gestational age, the use of human fat milk modular and 28 kcal/oz fortification were collected. Anthropometrics were assessed at 34 weeks, 36 weeks, and/or discharge. Z-scores were calculated using the 2012 Fenton growth curves via the online database PediTools. The database was utilized to plot all anthropometric measurements of individual patients [13]. This tool was used to report percentiles and z-scores with an integrated gestational age calculator. Secondary data collection was obtained through the Vermont Oxford Network database based on individual outcomes within Group 1 and Group 2. Supplemental data included length of stay, length of PN days, and malnutrition criteria.

### 2.4. Malnutrition Analysis

Per Goldberg et al., the primary indicators of neonatal malnutrition include a decline in weight for age z-score of 0.8–1.2 standard deviation (SD) for mild malnutrition, >1.2–2 SD in moderate malnutrition, and >2 SD in severe malnutrition [13].

### 2.5. Statistical Analysis

For categorical (qualitative) data, e.g., Vermont Oxford Index data or ethnicities, the comparison between the study groups used either the chi-square or Fisher’s exact tests. For the latter, an exact *p*-value calculation was used for tables with small expected frequencies (<5). For quantitative data, e.g., length of stay, the Wilcoxon rank-sum test was used. All statistical comparisons were performed at a 5% significance level.

## 3. Results

There were no significant differences in infant demographics or ethnicities between the groups (Table 3). Group 1 had a mean gestational age of 29.0 weeks, and Group 2 had a mean gestational age of 29.5 weeks (*p* = 0.34). The mean birth weight in Group 1 was 1.24 kg, and in Group 2, it was 1.28 kg (*p* = 0.6). A variety of ethnicities were represented in both groups. Group 1 included 30 infants, 16 males and 14 females. Group 2 included 36 infants, 22 males and 14 females.

Group 1 was fed an EHMD with the reactive use of a human milk fat modular. In Group 1, human milk +8 fortifier was used in 43.8% of the infants. Group 2 was fed an EHMD with the proactive use of a human milk fat modular. In Group 2, utilizing the updated feeding protocol with proactive human milk fat modular, the use of human milk +8 fortifier decreased to 13.9%. Group 1 required significantly more human milk +8 fortifier than Group 2 (*p* = 0.0075).

Both groups within the study utilized MOM as well as ready-to-feed donor milk-fortified products. There was no significant difference in the use of MOM between the groups (*p* = 0.47). In Group 1, 70% of infants had MOM fortified with Prolact +6. Of these infants, 43% required advancement to Prolact +8. Thirty percent (30%) of infants had no MOM and were fed with a Ready to Feed (RTF) 26 kcal/oz product. Of these infants, 44% required advancement to RTF 28 kcal/oz. All infants in Group 1 were fortified with the human milk fat modular prior to advancing calorie fortification. In Group 2, 78% of infants had MOM fortified with Prolact +6. Of these infants, 18% required advancement to Prolact +8. Twenty-two percent of infants had no MOM and were fed with RTF at 26 kcal/oz. Of these infants, none required advancement to RTF 28 kcal/oz. All infants in Group 2 were proactively fortified with human milk fat modular within the standard feeding protocol.

Utilizing neonatal malnutrition indicators [14], severe malnutrition declined from 3.3% to 2.7% from Group 1 to Group 2, respectively, while moderate malnutrition declined from 36.6% to 8.3% from Group 1 to Group 2. Overall, z-score weight change improved from birth to discharge within malnutrition severity (*p* = 0.0061) (Table 4).

The change in z-score birthweight to weight at 36 weeks improved for all infants in Group 2 (Table 3). Group 1’s mean change in z-score was −0.91 ± 0.42, and Group 2’s mean change in z-score was −0.80 ± 0.47 (*p* = 0.4). An overall improvement of 0.11 within the SD change was seen. The change in z-score birthweight to discharge weight improved by 0.17 SD (*p* = 0.41). The change in z-score birth length to discharge did not improve between Group 1 and Group 2. Of note, the median z-score length was much higher in Group 1 than in Group 2 (*p* = 0.43). The change in z-score head circumference to discharge improved by 0.21 SD (*p* = 0.53) in Group 2.

Weight change from birth to discharge improved within the two groups (*p* = 0.0061, Wilcoxon rank-sum test). In Group 1, 12 infants (40%) did not meet malnutrition criteria, with a difference in z-score for birthweight of 34 weeks. In Group 2, 17 infants (47.2%) did not meet malnutrition criteria, with a difference in z-score for birthweight of 34 weeks. The length change from birth to discharge was similar between the two groups (*p* = 0.91). In Group 1, 17 infants (56.6%) did not meet malnutrition criteria, with a difference in z-score for birth length to discharge. In Group 2, 18 infants (50%) did not meet malnutrition criteria with a difference in z-score for birth length to discharge. Head circumference, monitoring change in birth to discharge z-score, was stable within the two groups (*p* = 0.31). In Group 1, 19 infants (63.3%) did not meet malnutrition criteria with a difference in z-score for birth head circumference to discharge. In Group 2, 24 infants (66.6%) did not meet malnutrition criteria with a difference in z-score for birth head circumference to discharge. Of note, malnutrition was not an official ICD-10 code for billing purposes. Malnutrition indicators were utilized as a data collection standard.

### 3.1. Secondary Outcomes

Level III’s average length of stay (LOS) before an EHMD was established in 2014–2015 was 77.2 days (Table 5). The use of an EHMD diet initially led to a decrease of 2.2 days (averaging 75 days) in 2016–2017. Developing and revising the standardized feeding protocol, in addition to the use of a proactive fat modular in the standardized NICU feeding protocol, resulted in an even further decrease in LOS. Group 1 average LOS was 66.2 days, with a further decrease of 2.4 days (averaging 63.8 days) in Group 2. The average number of days on PN was similar between the groups (*p* = 0.73). Group 1 had an average of 8.3 ± 3.5, and Group 2 had an average of 8.2 ± 4.0.

Participation in the VON provided an additional review of co-morbidities between Group 1 and Group 2. The incidence of chronic lung disease decreased between Group 1 and Group 2. Overall, the percentage of infants with CLD decreased from 40% in Group 1 to 19.4% in Group 2 (*p* = 0.10).

### 3.2. Cost Analysis

All infants in Group 1 and Group 2 were assessed daily based on products utilized and feeding volume provided over a 24 h period. Based on the cost of fortification in 2022, the total cost of fortification in Group 1 was, on average, USD 14,748.13 per infant (*n* = 30). The total cost of fortification in Group 2 was, on average, USD 11,780.35 per infant (*n* = 36). Most infants were on multiple products while following the EHMD. Table 6 displays the use of each product, including how many infants utilized the specific product. The overall amount of product used and volume received were divided by 100 mL to obtain the number of bottles needed (10 mL per Prolact CR used). The total price was then divided by the number of infants utilizing that product to provide the resulting cost per infant on average. Based on this information, the total cost of fortification in Group 1 was USD 442,443.86, with an average of USD 14,748.13 per infant. The total cost of fortification in Group 2 was USD 424,092.42, with an average of USD 11,780.35 per infant. A total savings of USD 18,351.44 was seen in Group 2 (Table 6). The average cost saved per infant was USD 2967.78.

Based on the Prolacta Bioscience 2022 Price List, the list price of Prolact +6 is USD 193.13. With the addition of Prolact CR at the ratio of 4 mL per 100 mL feed, the total cost is USD 209.61. The list price of Prolact +8 is USD 257.50. With the addition of Prolact CR, the total cost is USD 273.98. The cost savings of utilizing MOM with Prolact +6 with the addition of Prolact CR vs. Prolact +8 with the addition of Prolact CR (at 4 mL/100 mL) is USD 64.37 per 100 mL. The list price of ready-to-feed (RTF) 26 kcal/oz is USD 206.00. With the addition of Prolact CR at the ratio of 4 mL per 100 mL fee, the total cost is USD 222.48. The list price of RTF 28 kcal/oz is USD 267.80. With the addition of Prolact CR, the total cost is USD 284.28. If a ready-to-feed product is required, the cost savings is USD 61.80 per 100 mL. The list price for 2022 will differ based on the institution’s purchasing price for fortification through contract negotiations.

Group 1 had an average length of stay of 66.2 days. The length of stay decreased to 63.8 days in Group 2, an average decrease of 2.4 days. Assuming a day of admission in the NICU costs an average of USD 3500 [15], the savings from the length of stay alone amounted to USD 8400 per infant.

## 4. Discussion

This quality improvement study displays significant cost savings with a reduction in the use of a +8 human milk fortifier by providing a human milk fat modular proactively within a standardized feeding protocol. An EHMD diet, the recommended nutrition source for all preterm infants, was utilized in both Group 1 and Group 2 within this study [16]. Recent studies have reported significant fat loss while utilizing MOM and donor human milk [8] feedings within this population. This study reviews the importance of a human milk fat module within a standard feeding protocol.

Target growth goals were maintained and improved in Group 2, as displayed by a change in z-score from birthweight to 36 weeks (*p* = 0.21) and birthweight to discharge (*p* = 0.41) (Table 3). The change in z-score from birth length to 36 weeks remained stable (*p* = 0.83). Group 2 also displayed improvements in head circumference z-score change from birth to 36 weeks (*p* = 0.82) and birth to discharge (*p* = 0.53).

Adhering to the standardized feeding protocol (Figure 3) assures all VLBW infants are meeting estimated nutrient needs, specifically for calories, protein, and fat, both enterally and parenterally. When a VLBW infant is receiving 160 mL/kg on average, the infant is being provided ~149 kcal/kg (varies due to the mother’s own milk as well as fat loss) and 4.32–4.48 gm/kg protein. Based on Koletzko’s latest recommendations [17], enteral protein is being met within these guidelines. Additional calories from fat are needed to support optimal growth. This quality improvement study showed improved growth and cost savings when a human milk fat modular was used earlier within a standardized feeding protocol.

Belfort and Ehrenkranz suggested that both greater weight gain, while supporting linear growth and head growth, and the use of human milk fortifiers can be associated with better neurodevelopmental outcomes [18]. Early fortification and proactive use of cream are two essential components of a successful feeding protocol utilizing an EHMD [1].

Limitations to this study include the use of a retrospective cohort study design with uncontrolled changes within the NICU during the two periods of data collection. The small sample size likely contributed to the lack of statistical significance. Excluded infants were removed from data collection if the feeding protocol was not being followed (Table 4).

## 5. Conclusions

The proactive use of a human milk fat modular within a standardized protocol for all VLBW infants demonstrated a significant reduction in the use of a high-calorie human milk fortifier, which in turn provided cost savings to the NICU. Proactive use of a human milk fat modular supported appropriate growth, trending toward improvement. Further cost savings were potentially found because of decreased co-morbidities and decreased length of stay. Further study is needed to confirm these findings in other institutions due to the small study size.

## Figures and Tables

**Figure 1 nutrients-16-01206-f001:**
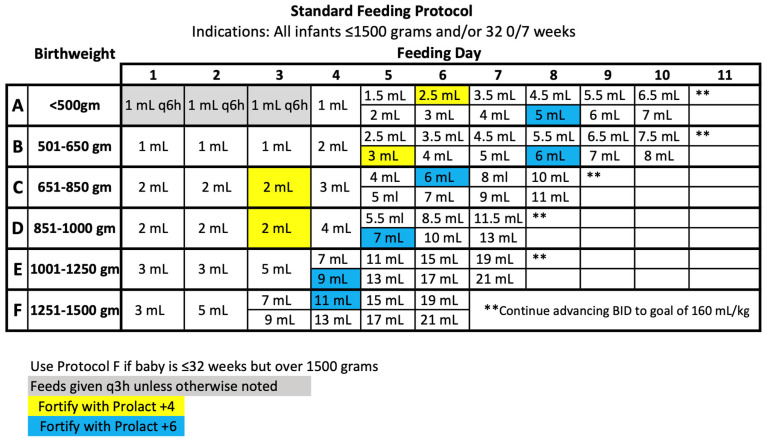
2020 Standard Feeding Protocol.

**Figure 2 nutrients-16-01206-f002:**
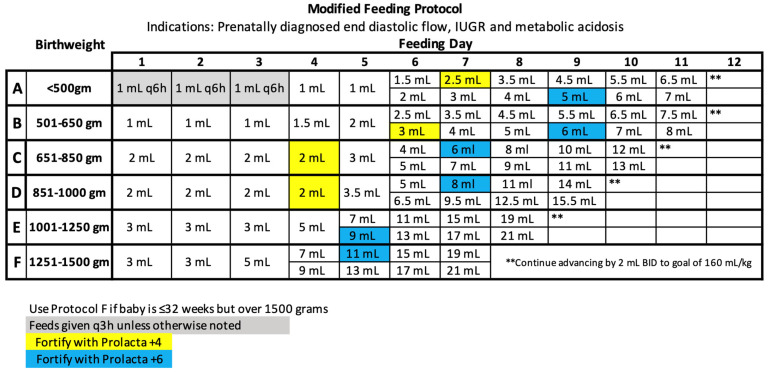
2020 Modified Feeding Protocol.

**Figure 3 nutrients-16-01206-f003:**
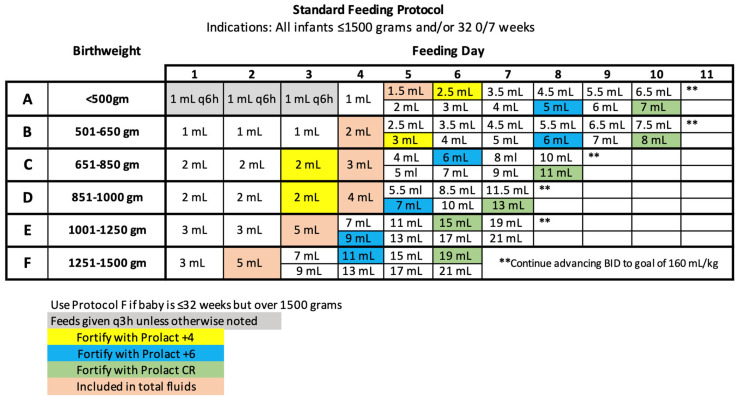
2021 Standard Feeding Protocol.

**Table 1 nutrients-16-01206-t001:** Malnutrition severity (birthweight to discharge weight), following Goldberg et al. indicators.

Severity	Group 1 *n* = 30	Group 2 *n* = 36
No Malnutrition	14	16
Mild Malnutrition	4	16
Moderate Malnutrition	11	3
Severe Malnutrition	1	1

*p* = 0.0061 (exact chi-square test).

**Table 2 nutrients-16-01206-t002:** Exclusions.

Exclusion Criteria	Group 1	Group 2
Presumed milk protein allergy	3	1
Nil per os (NPO)/Gastroschisis	1	--
Necrotizing enterocolitis	1	--
Spontaneous intestinal perforation	1	--
Early wean	--	1
Transport	1	1

**Table 3 nutrients-16-01206-t003:** Demographics/Ethnicities/Race.

Parameter	Group 1 *n* = 30	Group 2 *n* = 36	*p*-Value *
Sex (F)	14/30 (46.7%)	14/36 (38.9%)	0.52
Gestational age	29.0 ± 2.1 **	29.5 ± 1.9	0.34
Birthweight	1.24 ± 0.28	1.28 ± 0.28	0.60
Weight z-score birth	0.09 ± 0.76	−0.003 ± 0.86	0.82
Length z-score birth	−0.24 ± 0.93	−0.03 ± 1.05	0.27
Head circumference z-score birth	0.008 ± 0.99	−0.14 ± 0.89	0.64
Non-Hispanic White	13	13	0.55
Non-Hispanic Black or African American	5	8	0.57
Hispanic	9	7	0.32
Non-Hispanic Asian	0	4	0.12
Non-Hispanic, Other	3	4	1.0
Inborn	23	27	0.88

F = female; * sex, race/ethnicity, and location are analyzed by chi-square test; all others analyzed by Wilcoxon rank-sum test; ** mean ± SD.

**Table 4 nutrients-16-01206-t004:** Change in z-score.

Parameter	Group 1: Mean ± SD (Median)	Group 2: Mean ± SD (Median)	*p*-Value (Wilcoxon Rank-Sum Test)
Δ weight z: 36 weeks	−0.91 ± 0.63 (−0.88)	−0.76 ± 0.54 (−0.76)	0.21
Δ weight z: discharge	−0.94 ± 0.60 (−0.82)	−0.77 ± 0.57 (−0.81)	0.41
Δ length z: 36 weeks	−0.81 ± 0.93 (−0.70)	−0.82 ± 0.99 (−0.78)	0.83
Δ length z: discharge	−0.57 ± 1.01 (−0.63)	−0.71 ± 0.97 (−0.84)	0.43
Δ head circumference z: 36 weeks	−0.53 ± 0.85 (−0.56)	−0.48 ± 0.66 (−0.46)	0.82
Δ head circumference z: discharge	−0.44 ± 0.77 (−0.52)	−0.23 ± 0.98 (−0.38)	0.53

**Table 5 nutrients-16-01206-t005:** Secondary outcomes.

Parameter	Group 1	Group 2	*p*-Value *
Length of stay	66.3 ± 25.7 ** median = 61.5	63.9 ± 26.2 median = 63	0.83
Total parenteral nutrition (days)	8.3 ± 3.5 median = 7.5	8.2 ± 4.0 median = 7.0	0.73

* analyzed by Wilcoxon rank-sum test; ** mean ± SD.

**Table 6 nutrients-16-01206-t006:** Cost Analysis.

Product Used	Group 1 (*n* = 30)	Group 2 (*n* = 36)	Cost Difference
Prolact +6	USD 73,856.77	USD 55,775.69	−USD 18,081.08
# of Infants	21 * (USD *3516.99*)	28 * (USD *1991.99*)	
Prolact +6 w/Prolact CR	USD 95,833.69	USD 253,141.80	USD 157,308.11
# of infants	18 * (USD *5324.09*)	28 * (USD *9040.78*)	
Prolact +8 w/Prolact CR	USD 105,208.32	USD 47,541.01	−USD 57,667.31
# of infants	9 * (USD *11,689.81*)	5 * (USD *9508.20*)	
RTF 26 kcal/oz	USD 49,213.40	USD 4004.64	−USD 45,208.76
# of infants	9 * (USD *5468.16*)	8 * (USD *500.58*)	
RTF 26 kcal/oz w/Prolact CR	USD 34,867.07	USD 63,629.28	USD 28,762.21
# of infants	6 * (USD *5811.18*)	8 * (USD *7953.66*)	
RTF 28 kcal/oz w/Prolact CR	USD 83,464.61	--	−USD 83,464.61
# of infants	4 * (USD *20,866.15*)	0 (USD *0.00*)	
**Totals**	** USD **442,443.86**	** USD **424,092.42**	** −USD **18,351.44**

* Cost of product used per infant. ** Total cost of product used per group.

## Data Availability

The original contributions presented in the study are included in the article, further inquiries can be directed to the cooresponding author.

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
