# Peer review of "Proactive Use of a Human Milk Fat Modular in the Neonatal Intensive Care Unit: A Standardized Feeding Protocol"

_nutrients, 2024, doi:10.3390/nu16081206_

Round 1
Reviewer 1 Report
Comments and Suggestions for Authors
Dear authors,
The subject addressed in your manuscript is an important one and deserves to be studied. A standardized nutrition protocol is welcome for current neonatal practice, and breast milk is the ideal source of calories, protein, and fat. A particularly important aspect, especially in developing countries, is the cost, an additional reason for the implementation of these standardized protocols.
I would only have to mention the correction of some errors.
- Line 44 -46: please specify the meaning of VON, NEC and IUGR - at the first appearance in the text.
-Table 6 and the exclusion criteria should be moved to the Methods/Results section.
-The Discussion chapter should be developed by comparing with other similar studies.
Author Response
Please see attachment and updated manuscript.

Reviewer 2 Report
Comments and Suggestions for Authors
The manuscript entitled “ Proactive Use of a Human Milk Fat Modular in the Neonatal Intensive Care Unit: A Standardized Feeding Protocol” aims to demonstrate that With proactive use of a human milk fat modular in a standardized feeding protocol VLBW infants showed improved growth, lower malnutrition rates, and decreased use of higher caloric fortifiers.
The article is original, well written and methods are adequate. The references are up-to-date and mantaine approprite standards. Moreover, The text is well articulate and easy to read.
I suggest to deepen The discussion and provide more details about the participants.
Kind regards
Author Response
Please see attachment and revised manuscript.

Round 2
Reviewer 1 Report
Comments and Suggestions for Authors
Dear authors,
Thank you for your responses. The manuscript is improved. Only some minor corrections required: renumber the tables and the references.
Thank you.